# African Swine Fever: Fast and Furious or Slow and Steady?

**DOI:** 10.3390/v11090866

**Published:** 2019-09-17

**Authors:** Katja Schulz, Franz Josef Conraths, Sandra Blome, Christoph Staubach, Carola Sauter-Louis

**Affiliations:** Friedrich-Loeffler-Institut, Federal Research Institute for Animal Health, Südufer 10, 17493 Greifswald-Insel Riems, Germany; Franz.Conraths@fli.de (F.J.C.); Sandra.Blome@fli.de (S.B.); Christoph.Staubach@fli.de (C.S.); Carola.Sauter-Louis@fli.de (C.S.-L.)

**Keywords:** African swine fever, epidemiology, transmission, disease spread, mortality, case fatality ratio

## Abstract

Since the introduction of African swine fever (ASF) into Georgia in 2007, the disease has been spreading in an unprecedented way. Many countries that are still free from the disease fear the emergence of ASF in their territory either in domestic pigs or in wild boar. In the past, ASF was often described as being a highly contagious disease with mortality often up to 100%. However, the belief that the disease might enter a naïve population and rapidly affect the entire susceptible population needs to be critically reviewed. The current ASF epidemic in wild boar, but also the course of ASF within outbreaks in domestic pig holdings, suggest a constant, but relatively slow spread. Moreover, the results of several experimental and field studies support the impression that the spread of ASF is not always fast. ASF spread and its speed depend on various factors concerning the host, the virus, and also the environment. Many of these factors and their effects are not fully understood. For this review, we collated published information regarding the spreading speed of ASF and the factors that are deemed to influence the speed of ASF spread and tried to clarify some issues and open questions in this respect.

## 1. Introduction

The global concern regarding African swine fever (ASF) and the currently circulating ASF virus strains of the genotype II has substantially increased. This is because of the recent spread of the disease within Europe and Asia, where at least Cambodia, China, Mongolia, North Korea, Laos, the Philippines, and Vietnam have been affected so far. Owing to the constant spread of the disease, the expansion of areas in which ASF occurs at least in wild boar, and the increase in the number of affected countries, research and detailed epidemiological analyses are required to close knowledge gaps. By now, several important facts and assumptions are widely accepted, for example, regarding the main transmission pathways and the dominant role of human behavior in the spread of ASF [1,2,3,4,5]. However, there is still little reliable data, but controversial debate, on the transmission rate, the spreading speed of ASF, and the relevant epidemiological terms [6]. Understanding the spread and spatial distribution of a disease, however, is of utmost importance for disease prevention and control [7,8].

After the introduction of ASF into Georgia in 2007, it was initially hypothesized that ASF might either behave like a self-limiting disease and fade out quickly owing to its high mortality and case-fatality ratio or, on the contrary, that ASF might spread rapidly [1,9]. The real situation that emerged falsified both predictions. Currently, at least 14 countries outside Africa are affected by ASF, among them are several member states of the European Union (OIE WAHIS interface, visited online 26th July 2019). In August 2018, China, for which pig production and the consumption of pig products play a significant role, reported the first outbreaks of ASF [10,11]. Two months later, already 33 outbreaks were notified [12]. In May 2019, 134 ASF outbreaks were reported and, within a relatively short time, ASF has spread throughout a large area of the country, suggesting a rapid spread [13]. In addition, presumably through smuggled pork from China, ASF has been transmitted to pigs in other Asian countries including Cambodia, Mongolia, North Korea, Laos, the Philippines, and Vietnam. The epidemic that had its origin presumably in a port in Georgia in 2007 is far from being under control worldwide.

Many research publications and scientific opinions have so far characterized ASF as a highly contagious disease with high mortality [14,15,16,17,18,19]. However, recent research and the course of disease in affected countries, challenge the belief that ASF is a highly contagious disease [1,2,5,9]. ASF seems to be characterized by a high case-fatality ratio (i.e., most of infected animals die) paired with low to moderate mortality (only a limited proportion of the population becomes infected, at least in wild boar).

We aimed at collecting and summarizing published information on these uncertainties by searching for literature on transmission rates, spreading speed, and mortality in ASF outbreaks in commercial pig holdings, cases in wild boar, and in data from experimental studies.

## 2. ASF and Its Spreading Speed

The spread of a disease and its speed is very complex. It depends on many different characteristics. They not only include features of the pathogen, but also characteristics of the host, that is, of the susceptible pig or wild boar populations, and of the environment. We reviewed different, partly controversially discussed definitions and descriptions of epidemiological terms related to the spread of ASF, and refined some of these definitions. Thereby, we aimed at ensuring a uniform understanding of the presented terminology. In this process, we reduced the number of terms to those that may be considered as crucial for describing or explaining the spreading speed of ASF. Definitions of more general terms are included in the Appendix A.

### 2.1. Basic Reproductive Number (R_0_)

The basic reproductive number R_0_ is a measure of the ability of a disease to spread in a population. It is defined as the average number of secondary cases caused by one infectious individual during its entire infectious period in an entirely susceptible population, and constitutes an important quantitative property of an epidemic [20,21,22,23]. If the value of R_0_ is below 1, this indicates that the disease will disappear from the population, whereas values above 1 suggest that the disease will persist or spread within the population [23]. R_0_ is calculated by determining the product of the transmission probability, the average number of contacts, and the duration of infectiousness. This calculation of R_0_ is based on the assumption of a homogeneous mixing in the population, that is, all animals in the population are expected to have the same number of contacts [24]. Cross, et al. [25] pointed out that R_0_ may be a poor predictor of the course of disease and that other factors, such as transmission within groups or group size, are equally important for predicting the spread of a pathogen.

### 2.2. R_0_ in ASF Outbreak Situations

As mentioned before, R_0_ is highly dependent on a variety of factors. It depends, for example, on the infectiousness of the affected host, that is, the period it remains infectious, the number of susceptible individuals around the diseased animal, and the virulence of the circulating ASF virus (ASFV) strains or isolates [5].

One of the first published studies, in which R_0_ was estimated for ASF on the basis of field data for outbreaks in the Russian Federation, resulted in relatively high R_0_ values (8–11) for ASF infections in pig farms [18]. The R_0_ for transmission between farms determined in the same study was lower (2–3). In several field and experimental studies, similar values were obtained, whereby the R_0_ values for between-farm transmission were always lower than those for within-farm transmission [26,27] (Figure 1). The same applied for the R_0_ values in cases of indirect transmission [5] (Figure 2). By contrast, a very high R_0_ of 18.0 was determined in an experimental transmission study [28]. Yet, the confidence interval of this estimate was wide (6.90–46.9) (Figure 2), presumably owing to the limited number of experimental animals that can be included in such an experimental setting.

A summarizing analysis of published study results shows that the point estimates of R_0_ for ASF described in the literature so far cover a broad range from 0.5 to 18.0. This wide variation may inter alia depend on the properties of different virus isolates (e.g., virulence and infectivity), but also on the methods used to determine R_0_, which vary considerably [5]. However, regardless of the study setting (field or experimental), the calculated R_0_ values for within herd/direct transmission studies seem to be comparable. Likewise, R_0_ calculations for between herd/indirect transmission yielded similar values (Figure 1 and Figure 2). These results suggest that the studies included in this analysis obtained reasonably similar R_0_ values for ASF, despite the different methods and materials used. Unfortunately, we were unable to investigate whether publication bias may have affected the outcome of this analysis, because most of the available original publications did not contain the required raw data on the size of the study population.

Comparing the R_0_ of ASF with R_0_ values of other pig diseases, it was found that studies calculating the R_0_ of foot and mouth disease, which is described to be highly contagious, often resulted in values higher than 20 [29,30,31]. Classical swine fever is also described as a highly contagious disease. In several studies, the R_0_ showed values similar to that of ASF [32,33]. As in ASF, the values differed between within and between pen infections. In the study of Klinkenberg, et al. [34], the within-pen-R_0_ in weaner pigs was as high as 100 (95 percent confidence interval (CI) 54.4–186). However, also in the case of these two diseases, the R_0_ was highly dependent on several other factors, like study design, virus strain, and so on.

### 2.3. Spatio-Temporal Distribution and Disease Spread

The character of transmission and spread of disease cannot be viewed independently of spatial and spatio-temporal vicinity. If individuals that are at risk live close to each other, transmission is more likely as compared to a situation where they are separated through a great distance [7]. The spatial distribution is often only visualized using maps. However, there are several analytical tools such as spatial cluster detection or spatio-temporal modelling, which can be used to include sound spatio-temporal analyses into epidemiological investigations [7,8,35]. Several studies are published describing the use of different models to investigate the spatio-temporal trend of animal diseases [36,37,38,39,40,41]. Moreover, simulation models are widely used to describe the spatial and temporal spread of animal diseases. They can also be used to analyze patterns of endemic and epidemic diseases or to evaluate the success of control measures [42,43,44,45,46,47,48,49,50,51].

### 2.4. Spatio-Temporal Distribution and Spread of ASF

In this section, we tried to summarize the results of publications that focused mainly on the spreading speed of ASF. The spreading speed highly depends on several factors that were already discussed or will be discussed later; therefore, some of the studies dealing with the speed of ASF spread or transmission rates are included in discussions further above or below.

Modelling the spatial spread of ASF in a wild boar population, Fekede, et al. [52] showed that seasons with temperatures lower than 0 °C support the occurrence of ASF and, therefore, the spread of the disease. In several other studies, an increased detection of ASF-positive wild boar in wintertime was confirmed [3,53,54]. In contrast to wild boar, it was found that the majority of ASF outbreaks in domestic pigs were detected in summer [53,54,55,56].

The spatial spread of ASF within the wild boar population in Poland was described as slow [55,57]. Podgorski, et al. [58] hypothesized that this slow spread could be because of the spatial constraints on contacts between wild boar. In their cluster analysis of ASF cases in Russian wild boar, Iglesias, et al. [59] found that individual wild boar cases were spatially associated over a radius of up to 130.79 km and within a maximum of 90 days. The observed spreading patterns were explained through direct and indirect wild boar contact, but may have also been caused by the potential spread of infected wild boar due to increased hunting. In a similar study from Sardinia, Iglesias, Rodriguez, Feliziani, Rolesu and de la Torre [54] identified a maximum spatial association of 25 km between ASF cases in wild boar with an assumed daily movement distance of 2–10 km. In the same study, the maximum distance of spatial association between ASF notifications in domestic pig holdings was 15 km [54]. In a simulation model study, the spreading distance between herds was estimated to be lower than 10 km [60]. However, in the Russian Federation, Korennoy, et al. [61] calculated a mean distance of 156 km between two connected domestic pig outbreaks and an average time period of 7.5 days. Following the statements of the Russian veterinary authorities, Blome, et al. [62] reported already in 2011 about a spreading speed of 350 km per year of ASF within the Russian Federation. These results suggest a rapid spread of the disease. However, in the study of Oļševskis, Guberti, Serzants, Westergaard, Gallardo, Rodze and Depner [2], a slow spread of ASFV within infected pig herds was described. Also, Nurmoja, Mõtus, Kristian, Niine, Schulz, Depner and Viltrop [56] reported a rather slow spread within affected pig herds, suggesting a relatively low infectiousness.

The general terminology for the following terms can be found in the Appendix A.

### 2.5. Morbidity in ASF Outbreak Situations

Morbidity can be used to characterize the speed of spread of ASF. High morbidity within a defined period requires a certain level of infectivity and a relatively high virulence of the agent. We thus hypothesize that a disease that causes high morbidity is likely to spread with high speed within the target population.

Owing to the high case fatality ratio of ASF, its morbidity values are expected to resemble those for mortality. Infections with a pathogen of low or moderate virulence are likely to result in a higher morbidity, but lower mortality. Therefore, high morbidity was mostly described in experimental studies, which also demonstrated a high mortality [27,63,64,65]. However, in a few studies, animals that had developed clinical signs after infection, recovered completely, thus suggesting a higher morbidity than the observed mortality [66,67,68].

### 2.6. Mortality and Case Fatality Ratio in ASF

Mortality and case fatality ratio are parameters that provide information on circulating ASF viruses and on disease dynamics. However, when interpreting the study results, it should be kept in mind that both terms are frequently confused or incorrectly used as synonyms.

A fast spread of ASF is not necessarily determined by a high case-fatality ratio, as the latter term refers only to the proportion of cases of death among the diseased animals. Likewise, high mortality, which in contrast defines the proportion of death animals among the whole population at risk, does not necessarily correlate with a high speed of ASF spread. Therefore, both terms should always be considered in the context of other parameters such as host-related factors (e.g., age) and properties of the agent (e.g., virulence of the circulating strains). Recent studies showed that the mortality of ASF was lower in older pigs and in animals infected with low-virulent strains [69,70]. Blome, et al. [71] described that the mortality of ASFV ranged from 3% to 100%, depending on the virulence of the virus strain. Mebus [72] found different levels of virulence of ASFV isolates and corresponding values of mortality (from less than 20% to 100%). An experiment with several moderately virulent ASFV isolates led with almost all used strains to moderate mortality. Only the Brazil’78 isolate, though described as moderately virulent, killed all infected pigs, that is, it caused 100% mortality in this experiment [70]. The fact that mortality depends on the virulence of particular ASFV strains or isolates was also confirmed by Vlasova, Varentsova, Shevchenko, Zhukov, Remyga, Gavrilova, Puzankova, Shevtsov, Zinyakov and Gruzdev [64].

Montgomery [73], who first described ASF, found a very high mortality of the disease, both in ASF outbreaks and in experimental infections.

Under the assumption that ASF usually has an extremely high mortality, syndromic surveillance of pig mortality has been proposed as the most appropriate surveillance method for ASF in domestic pig holdings [74]. In the experimental trial of Gallardo, et al. [75], mortality and case fatality ratio both reached the high value of 94.5% when domestic pigs were inoculated with the ASFV Lithuania 2014 genotype II field isolate. Pietschmann, Guinat, Beer, Pronin, Tauscher, Petrov, Keil and Blome [27] used another isolate of genotype II and observed 100% mortality in wild boar (within 17 days post infection) and domestic pigs (within 36 days post infection). Experimental exposure to a highly virulent Caucasian ASF isolate (2008 isolate from Armenia) resulted in 100% mortality [76]. In this study, transmission among wild boar was faster than among domestic pigs. In a subsequent study, Blome, et al. [77] also found 100% mortality in wild boar infected with an ASFV Caucasus isolate. Using the same isolate, Guinat, Reis, Netherton, Goatley, Pfeiffer and Dixon [65] obtained 100% mortality in a trial using domestic pigs. Most of the experimental studies reviewed here are also summarized in a Scientific Opinion of the European Food Safety Authority [78].

Similar to the results of experimental infections, the mortality of ASF was found to be high in field studies. The course of an ASF outbreak in Malta in 1978 indicated a fast spread with a high mortality. The epidemic resulted in the loss of two-thirds of the pig population of Malta. It was finally decided to slaughter all the remaining pigs on the island [79]. The high virulence of the isolate (Malta/78) was confirmed in an experimental study, which revealed a mortality of 93.3% [80]. 

When data from ASF outbreaks in Nigeria in 2001 were analyzed, it became evident that the disease had spread very quickly. Although the mortality varied depending on the age of the pigs, it was not lower than 75.9% in any of the age groups [81]. In outbreaks that occurred in Tanzania in 2003 and 2004, mortalities of 82% and 72%, respectively, were recorded [82].

In 2014, Estonia was confronted with the first ASF cases in wild boar. A high mortality was observed in wild boar in the southern part of Estonia, whereas low mortality was found in the northeast of the country [68]. Yet, experimental infections carried out to determine the virulence of the strain that circulated in the northeast of the country showed that the mortality of the isolate was still very high, because 9 of the 10 inoculated wild boar died from ASF infection or became so ill that they had to be euthanized [68].

Recent findings suggest that it may take up to one month until an ASF introduction in a pig herd is detected because of increased mortality, thus indicating a rather moderate mortality for ASFV strains currently circulating in the respective region, in this particular case, in the Russian Federation [6]. A low mortality in the course of an ASF epidemic had already been observed; using an ASF isolate from Cameroon, which had caused high mortality in the field, Ekue, et al. [83] found a very low mortality in their animal experiments. Studies analyzing ASF outbreaks in Malawi suggested a mortality that was significantly lower than 100% in most areas [84].

In outbreaks that occurred in Belgium in 1985, the spread of ASF was described as slow, not only from one pen to another, but even within the same pen. Also, the mortality of the circulating ASF isolate or strain was apparently moderate [85].

When summarizing the results of the literature search regarding mortality, it seems obvious that studies prevail that describe a high mortality of circulating ASFV isolates or strains. It also became apparent that mortality is highly dependent on the virulence of the circulating virus isolate or strain. Moreover, similarly to most pathogens, the properties of the affected host species (in particular age, but also health and feeding condition) seem to play an important role. Therefore, ASF outbreaks are usually, but not always, characterized by a high mortality.

### 2.7. Infectiousness and Latent Period of ASF 

Under the specific conditions of ASF, infectiousness describes the period in which an infected animal can transmit the disease. The longer this period lasts, the higher the risk of transmission and, consequently, the risk of ASF spread increases accordingly. The period of infectiousness is highly dependent on the virus strain or isolate. This could be shown in an extensive transmission study, in which a minimum infectious period of 6–7 days was determined for ASF, while the maximum infectious period ranged from 20 to 40 days [28]. The study also showed that the infectiousness also depended on the transmission rates and pathways. In an earlier study, the authors had detected ASFV genome by polymerase chain reaction (PCR) in blood and oropharyngeal fluid even until 70 days post-infection, that is, until the end of the observation period [70]. In addition, infectiousness is defined by the amount of pathogen that is excreted by the animal [20]. Several studies showed that the amount of ASFV is clearly higher in blood then in other excretions [64,65,70,86].

The infectiousness of a pig suffering from hemorrhagic diarrhea due to ASF infection is higher than that of an animal showing only fever or loss of appetite. This difference may be explained by the fact that ASF is extremely efficiently spread through the blood of infected animals. In several transmission studies, which were conducted by Guinat, Gogin, Blome, Keil, Pollin, Pfeiffer and Dixon [5] in domestic pigs and in wild boar, the infectious period lasted from 2 to 14 days, while the latent period of ASFV was found to be 3–6 days. Similar results were obtained by Guinat, Porphyre, Gogin, Dixon, Pfeiffer and Gubbins [6], where the infectious period ranged from 4.5 to 8.3 days and the latent period from 5.8 to 9.7 days. In the study of Gulenkin, Korennoy, Karaulov and Dudnikov [18], the latent period was 15 days. Differences in the estimates may inter alia be the result of variations in the experimental design, different properties of the ASF virus isolates or strains, and the pigs or wild boar used in these studies.

Looking at infectiousness in connection with spreading speed, the ongoing discussion regarding potential ASFV carriers must not be ignored. Sanchez-Vizcaino, et al. [87] stated that animals can develop a chronic form as ASF and become carriers that can infect conspecifics even after a long period. In the study of Gallardo, Soler, Nieto, Cano, Pelayo, Sanchez, Pridotkas, Fernandez-Pinero, Briones and Arias [75], one pig that had contact with infected pigs survived the infection, and it remains unclear if such an animal has the potential to spread ASF. However, in other studies, it was observed that no disease transmission from surviving animals and contact animals took place, suggesting that survivors might play a negligible role in the spread of ASF [66,67,68].

With regard to the speed of spread, not only the duration of the infectious period is important, but also that of the incubation and latent period [60,88]. When transmission can take place before any clinical signs occur, that is, during the time when the incubation period and the infectious period overlap and the latent period is accordingly shorter than the incubation period, the spread of disease may be faster, as it is unlikely that control measures are applied within this period [88]. However, this applies only to domestic pigs, as the onset of clinical signs in wild boar is difficult to observe, in particular after the initial entry of ASF into a wild boar population.

### 2.8. Infectivity of ASFV

The infectivity of ASFV, usually measured as the 50% lethal dose (LD50) in tissue culture (TCID50), also plays an important role in determining the speed of spread of ASF.

It can be assumed that a virus with low infectivity will spread more slowly than a pathogen with high infectivity. However, the infectivity of ASFV is highly dependent on properties of the virus isolate or strain, in particular on its virulence, on the medium (blood, urine, other body fluids, feces, tissue, and so on), on any potential processing of this material (temperature, pH, storage period, and so on), and on the route of transmission. As already mentioned, transmission through direct blood contact is the most efficient route [27,63,76]. Olesen, Lohse, Boklund, Halasa, Belsham, Rasmussen and Botner [86] showed that the infectivity of ASFV in the environment of the studied pen was low. Correspondingly, transmission through other fomites like urine, feces, or feed appears less effective as compared with direct exposure to the blood of infected animals. However, in the case of hemorrhagic enteritis, feces can play an important role in the transmission of ASF. These assumptions are supported by several studies, which found higher PCR titers in blood than in other excretions [64,65,70]. Direct contact of an uninfected pig to the blood of another pig can be considered as less likely than contact to other excretions (in particular feces and urine) or ASFV-contaminated soil. Thus, the spread of ASF may be slower than that of a disease, where large amounts of infective virus are present in the environment.

### 2.9. Contagiousness of ASF

Regarding the contagiousness of ASF, no information was found in the literature that could add knowledge to that already described in relation to infectiousness.

### 2.10. Virulence of ASFV

Virulence describes, like infectivity, a property of a pathogen. A virus with a moderate or low virulence can still be highly infective [89]. However, it is undisputed that there is a relationship between virulence and infectivity. The effectiveness of different transmission routes might also be influenced by the virulence of a virus strain or isolate. The transmission of ASF was found to be more efficient when a highly virulent ASFV strain was used [90]. McVicar [91] found that the amount of virus in oronasal fluid was higher in pigs infected with highly virulent ASFV strains as compared with infections with an isolate of lower virulence. Thus, the virulence of the circulating ASFV strain has an effect on the spread of ASF and on the speed of spread. By contrast, Guinat, Gogin, Blome, Keil, Pollin, Pfeiffer and Dixon [5], who summarized the results of several studies, found that the virus load in different body fluids was very similar, regardless of the virulence of the virus strain or isolate used to infect the pigs.

Depending on the virulence of the virus strain or isolate, infection with ASFV can result in only mild clinical symptoms or 100% mortality or anything in between (Table 1).

The examples in Table 1 illustrate the complexity and potential inter-dependency of several parameters regarding virulence and the course of field and experimental infection with different ASFV isolates or strains. When comparing the presented results, the different infection routes should not be neglected (Table 1). However, the course of an ASF epidemic in wild boar may last for a long time, regardless of the virulence of the circulating strain [1,92,93]. In Brazil, where the virus strain circulating in domestic pigs was characterized as low virulent, it took several years until the disease had been eradicated [94]. In this case, the epidemic spread over the entire country and also affected several neighboring countries.

### 2.11. Tenacity or Resistance of ASFV to Environmental Factors

Several studies have shown that ASFV is resistant to extremely harsh environmental conditions, and thus can stay infectious for a long time in various matrices. The tenacity of ASFV is particularly high at cold temperatures, for example, in frozen meat, where the virus may stay infectious for several years [16,97,98]. Even at room temperature, substantial infectivity is preserved for months. Dee, et al. [99] found that ASFV stayed infectious for a few days in different feeds and feed ingredients, for example, in moist dog and cat food. It is also known that ASFV can survive putrefaction [78]. Mebus, et al. [100] found a resistance of ASFV across pH-levels ranging from 4 to 13.

## 3. Discussion

The view that ASF represents a highly contagious disease, spreading very fast and killing all pigs of an affected farm or the whole wild boar population in a region, requires substantial revision. We aimed to clarify a number of aspects in relation to factors that may be important in this respect, particularly those affecting the speed of ASF spread. We identified and described terms that are likely to play a role in the spread of ASF and searched for published information regarding these parameters.

Our findings emphasize the difficulty to define the speed of spread and the parameters relating to it. This is not only because of different definitions of the parameters that may influence the speed of spread, but may also be related to the interdependencies of many parameters on (other) properties of the agent and those related to the host or the environment. A further drawback of the current review is the fact that relatively few, and even less reproduced (and thus presumably reliable), studies concerning the specified parameters were available.

The different properties of the virus isolates and strains, but also the characteristics of the host factors, environmental parameters, and matrix effects, are difficult to separate from each other. Properties are often prone to influence each other and sometimes the definitions for the studied parameters collide if different studies are compared. Furthermore, it appears that some terms are not clearly defined or are used in various, sometime confusing ways in different studies. This refers mainly to the terms mortality and case-fatality ratio, but virulence, infectiousness, and infectivity may also be affected. With regard to mortality and the case-fatality ratio, the size of the susceptible population may be difficult to determine, particularly in wild boar, even if it is defined as the population living in a certain area or belonging to a particular pack of wild boar. On the basis of these parameters, it is often difficult to draw reliable conclusions concerning the true speed of ASF spread.

Depending on the ASFV strain, infection led quickly to 100% mortality, indicating a fast spread within separate epidemiological units [25,57,60,69,93]. Despite the high virulence of the ASFV isolates used for experimental infections and the corresponding high mortality observed, the current course of ASF in Eastern and Central Europe indicates a rather slow spread in the wild boar population over time. The largest amount of ASFV can be found in the blood, which makes direct transmission through exposure to blood of infected animals the most effective way of infection [57,60,62]. However, in other body fluids of infected pigs or wild boar and in the environment, the amount of infective virus is much lower [56,78]. As direct exposure to blood of infected animals is less likely than exposure to other body fluids or contaminated materials in the environment, this may reduce the morbidity, and thus also the speed of ASF spread. However, this might explain the differences in mortality between experimental and field settings, as direct contact to blood is more likely in experimental settings than in the field. Until now, the role of chronically infected animals within the spread of ASF in wild boar is highly disputed [58,59,60,94]. If animals that have recovered from ASF were still able to spread the disease, this could certainly influence the spreading of ASF and its speed.

Another potential hypothesis regarding the slow spread in the field might be a relatively low virulence of the ASFV circulating in the area. However, for Eastern and Central Europe, this hypothesis is not warranted by the results obtained in experimental studies using the strain circulating in this area, which suggest a high virulence of this ASFV strain [25,56,57,58,60,69,95].

Moreover, it seems puzzling that ASF is not self-limiting, in particular in view of the difficulties in transmission described above. Some studies demonstrated that ASFV stays infectious for a long time in various tissues [90,91,92,96,97,98,99]. Therefore, it can be assumed that the presence of infectious ASFV represents a risk of exposure of naïve animals and that this risk may be cumulating over time, even if the risk of exposure is relatively low at any given point of time. Correspondingly, the tenacity of ASFV seems to play a major role in the spread of ASF.

In countries where the wild boar population is heavily affected in large areas, the disease has often spread continuously and has led to a significant reduction of the wild boar population [29,85,100]. It is thus important to note that there are several additional factors influencing the spreading speed of ASF in addition to those listed in this review. Several studies mention the population density of wild boar, the density of infected wild boar in the proximity of commercial pig holdings, and the density of pig holdings in a defined area as risk factors for a faster spread of ASF [2,15,48,53]. However, the effect of the population density on the spread of ASF is still disputed, and no population density threshold could be defined so far to stop ASF spread [29,47,84].

In addition, the calculation or estimation of transmission speed is hampered by the behavior and the living habits of wild boar. Wild boar usually live in a pack with regular interactions within the pack, but not between different packs [50]. Although Pfeiffer, Robinson, Stevenson, Stevens, Rogers and Clements [7] stated that transmission is more likely when many animals live closely together, the current knowledge on the behavior of wild boar has to be incorporated into any assessment of ASF disease spread in the field.

When discussing the transmission speed of ASF, the different transmission cycles have to be taken into account [4,101]. Blood of infected pigs is the most efficient medium for ASF transmission [71]. This fact affects every transmission cycle.

The domestic cycle in the form of animal movement patterns, both within and between herds, plays a major role in the spread of ASF among domestic pigs [52,101]. This fact and others suggest, in turn, that biosecurity measures can limit the spread of ASF [2,3]. The hypothesis that high biosecurity standards can decelerate or even prevent the spread of ASF is supported by an increased number of ASF outbreaks in backyard farms with less than 50 pigs, and usually lower biosecurity measures [2,15,102]. Therefore, it can be assumed that the spread of ASF is faster in countries with a high number of small private pig holdings, as can currently be seen in Romania [103]. However, Estonian researchers could show that the biosecurity level had no measurable influence on the risk of an introduction of ASF [56]. Also, in China, it seems that the size of pig holdings has no major influence on the speed of spread of ASF. The number of backyard pig holdings has clearly declined in recent years in China. Despite that households in rural areas keeping pigs do not exceed 20%, a fast spread of ASF throughout the country has been observed [13,52].

Soft ticks of the genus Ornithodoros can also play an important role in the transmission of ASF. This is undoubtedly the case in the tick–pig cycle and in the natural sylvatic cycle of ASF in warthogs in southern Africa, but may also be relevant in other regions, if Ornithodoros ticks are present [5,102] so that it must be expected that these tick vectors can influence the speed of ASF spread. However, there is currently no evidence that ticks play a role in the ASF transmission cycle in Europe [103].

Finally, as mentioned in the introduction, it is of utmost importance to stress that human behavior is without any doubt the most important factor that can facilitate the transmission of ASF over long distances [2,5,47,88,104,105]. In countries with many private backyard holdings, the spread of ASF can be supported by failure of reporting outbreaks or suspect cases and, in the worst case, even by selling sick animals to circumvent problems expected by pig holders if ASF is detected. The influence of the human factor also becomes evident when the introduction of ASF into wild boar in the Czech Republic in 2017 and Belgium in 2018 is taken into consideration. Before these countries were affected, the nearest outbreaks of ASF had been hundreds of kilometers away. Therefore, similar to the foot-and-mouth epidemic in England in 2001 [27], the spatial pattern of disease outbreaks can often not only be explained through distance-dependent transmission, that is, through infected animals in close proximity to each other, but also through the transportation of infected animals or contaminated material over long distances. It is obvious that in scenarios where ASFV is introduced by distance-independent mechanisms, for example, transportation of infectious material in the course of various human activities, the parameters discussed in the context of this review play no or only a minor role.

## 4. Conclusions

On the basis of the available literature, we propose revising the view that ASF generally has to be referred to as a highly contagious disease. We tried to show that it is not always easy to answer the question raised in the title of this review, because the answer may depend on several epidemiological parameters. ASF is neither generally fast and furious nor is it slow and steady, but the appearance of ASF can be diverse. ASFV strains can vary in their virulence. However, highly virulent strains or isolates, also the one currently circulating in Eastern and Central Europe, which has recently been introduced into China, may be characterized by a low morbidity potentially owing to transmission through materials with a relatively low virus load, leading to slow spread in wild boar populations. Jumps of ASF over long distances are usually the result of human activities, and are thus unpredictable.

## Figures and Tables

**Figure 1 viruses-11-00866-f001:**
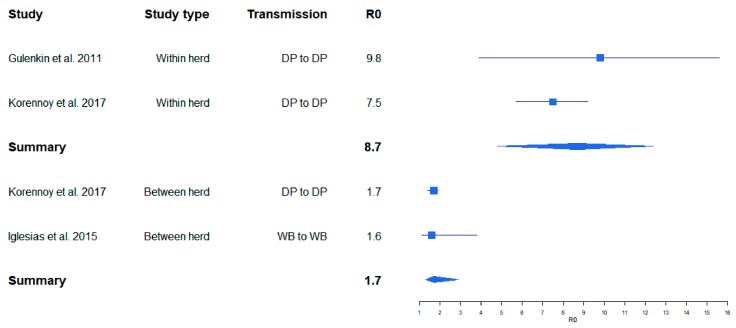
Variation of calculated R_0_ for African swine fever (ASF) obtained from ASF field studies. Boxes illustrate the calculated R_0_. The lines illustrate the confidence intervals. DP = domestic pig, WB = wild boar.

**Figure 2 viruses-11-00866-f002:**
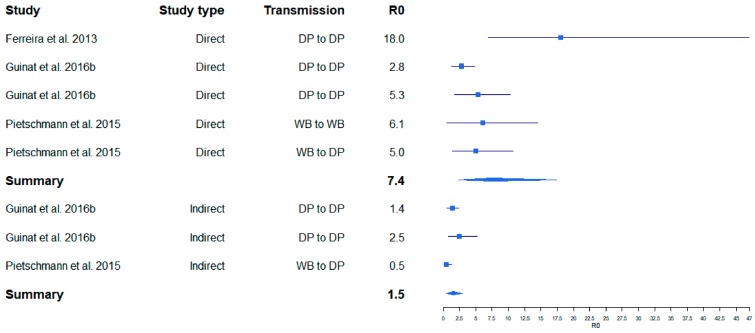
Variation of calculated R_0_ for ASF obtained from ASF experimental studies. Boxes illustrate the calculated R_0_. The lines illustrate the confidence intervals. DP = domestic pig, WB = wild boar.

**Table 1 viruses-11-00866-t001:** Strains and isolates of African swine fever (ASF), their virulence, and course of field and experimental infections.

ASFV Strain or Isolate	Virulence	Course of Disease	
		Field Infection	Experimental Infection/*Route of Infection*
Malta isolate (Malta/78)	Moderate/high	Fast spread in the domestic pig population, which ended in the slaughter of all pigs in Malta within one year [79].	Mild clinical signs in less than 50% of infected pigs, high mortality/*exposure to infected donor pigs* [80].
Brazil’78	Low/moderate	Mild clinical course and decreasing mortality over time in domestic pigs. Wide distribution throughout Brazil for at least eight years [94].	High mortality/*intranasally* [70].
Netherlands’86	Moderate	No information found	Low mortality with a rather subclinical course of disease/*oronasally and through contact* [67,95].
Georgia 2007/1	High	Large scale epidemic, wild boar and domestic pigs affected [96].	Moderate clinical signs, high mortality/*intramuscularly* [65].
Estonian’15/WB, Tartu-14	High	Rapid spread in the wild boar population [37].	100% mortality in experimentally infected domestic pigs. Two survivors in in-contact pigs/*intramuscularly* [66].
Estonian-Ida Viru	High	Only local spread within the wild boar population [37].	Almost 100% mortality (one survivor)/*oronasally* [68].
*Armenia’08*	*High*	*No information found.*	100% mortality in wild boar and domestic pigs/*oronasally* [27].

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
