# Peer review of "African Swine Fever: Fast and Furious or Slow and Steady?"

_viruses, 2019, doi:10.3390/v11090866_

Round 1

Reviewer 1 Report

The manuscript is covering an important hot topic and has a good value in terms of summarizing the published literature related to describing the transmission dynamics of ASF. However, I think the authors should spend some more effort on discussing the transmission cycle of ASF between domestic pigs, wild suids and ticks, as it is critical for shaping the epidemic progression of the disease. Furthermore, the authors need to make some comparisons on the transmission dynamics between ASF and other common swine diseases in Europe to be able to make a proper conclusion about the speed of disease spread.

Author Response

Reply to Reviewer

We agree with the reviewer that the transmission cycles play a role in the transmission route. However, in the introduction (lines 32-34), we emphasize that the present review acknowledges the existing and published knowledge about the different transmission cycles of ASF. However, the focus of this review was put on knowledge gaps regarding the spreading speed of ASF. To highlight the importance of the different cycles for the spreading speed, we added more information in the discussion (lines 387 - 403).

Regarding the comparison with other diseases, we added additional information to the R0 part (lines 109-116). However, as the speed of spread for other diseases is also very complex, it is beyond of the scope of the presented study to review all discussed terms for a variety of diseases.

Reviewer 2 Report

The review article “African swine fever: Fast and furious or slow and steady?” compares all published information about the infectivity of different ASFV strains. Including the current outbreak strain in Eurasia. It has been well thought out, and well written, and offers a nice insight that although in todays world the perception is fast and furious as the authors call it, due to the highly virulent Eurasia strain plaguing Asia and Europe.  There are strains of ASFV that don’t spread so quickly, and it is possible for introduction of one of these strains into an uninfected area, as well as these strains starting to occur over time (if not controlled) in current outbreak areas.

1.       The authors touch upon it at the end of the review that the current  strain has low morbidity and due to the infectious dose being low it was able to be spread rather quickly though materials.  However the authors don’t mention anything about the speed this strain spread over china, and how it has been spreading to neighboring countries. They also touch upon that that human factors are impossible to predict. This should also be elaborated on, to mention that lack of reporting, or small farmers selling/transporting animals once they become sick, continue to increase the spread of the disease. There should be a small section better describing the current outbreak’s across Eurasia.

2.       In the review the authors compare the virulence of several of the strains of ASFV, were all of the testing conditions the same in the referenced manuscripts.  For example using an infected animal with 10 ID50 to infect other animals only by shedding is much different than inoculating every animal. Could some of the differences described be due to these limited studies and limited details.  The mode of inoculation to access virulence should be listed in the table.

3.       The authors also don’t mention the potential impact of different farming procedures in different countries. For example backyard farms vs overcrowded production facilities.

4.       The review also doesn’t mention that if an area cannot control ASFV, over time additional variants or strains can occur such as what was seen in Estonia or what has been seen in other endemic areas, it is possible and highly likely that less virulent strains could arise in Asia, and these strains could continue to cause sporadic disease, or sporadic outbreaks not showing the same phenotype as is seen during the current outbreak strain.

Author Response

We added more information about the spread in China (lines 45-49), the potential role of backyard holdings (lines 391-400) and the ’human factor’ (lines 408-411). In addition, we added references in which these factors are discussed in more detail. A more detailed description of the current epidemic in Europe is beyond the scope of the study. We already mentioned the rather slow spread of the disease (line 351).  We added the inoculation routes in the table and added a sentence accordingly (line 308-309). We added more information (lines 394-402). We discussed the dependencies of the different characteristics from the different ASFV strains in each appropriate section and in Table 1. However, the co-existence of different strains in various countries does not influence directly the speed of spread of the ASF. The potential emergence of less virulent strains due to long circulation of the virus still remains a hypothesis, which is why we decided not to include this discussion in the presented paper.